# Preparation and Characterization of Bio-Nanocomposites Film of Chitosan and Montmorillonite Incorporated with Ginger Essential Oil and Its Application in Chilled Beef Preservation

**DOI:** 10.3390/antibiotics10070796

**Published:** 2021-06-30

**Authors:** Yin-Ping Zhang, Xin Wang, Yi Shen, Kiran Thakur, Jian-Guo Zhang, Fei Hu, Zhao-Jun Wei

**Affiliations:** 1Anhui Academy of Agricultural Sciences, Crop Research Institute, Hefei 230031, China; zhangyinping66@163.com; 2School of Food and Biological Engineering, Hefei University of Technology, Hefei 230601, China; 18158951863@163.com (X.W.); 2019111359@mail.hfut.edu.cn (Y.S.); kumarikiran@hfut.edu.cn (K.T.); zhangjianguo@hfut.edu.cn (J.-G.Z.); 3School of Biological Science and Engineering, Collaborative Innovation Center for Food Production and Safety, North Minzu University, Yinchuan 750021, China

**Keywords:** ginger essential oil, bio-nanocomposite films, antibacterial activity, chilled beef, shelf life, food preservation

## Abstract

In this study, bio-nanocomposite films containing different proportions of ginger essential oil (GEO), chitosan (Ch), and montmorillonite (MMT) were prepared and characterized, and the antibacterial effect of bio-nanocomposite films on chilled beef was evaluated. Fourier transform infrared analysis showed a series of intense interactions among the components of the bio-nanocomposite films. The infiltration of GEO increased the thickness of the film, reduced the tensile strength of the film, and increased the percentage of breaking elongation and the water vapor permeability. The migration of phenols in the films began to increase exponentially and reached equilibrium at about 48 h. The bio-nanocomposite films (Ch +0.5% GEO group, and Ch + MMT + 0.5% GEO group) effectively delayed the rise of pH, hue angle, and moisture values of chilled beef with time and slowed down the lipid oxidation and the growth of surface microorganisms on chilled beef. Altogether, the prepared biological nanocomposites can be used as promising materials to replace commercial and non-degradable plastic films.

## 1. Introduction

Beef is the third most consumed meat in the world, preceded by pork and poultry. China, United States, and Brazil are the world’s top three consumers of beef [1]. Beef is popular among the public and some fitness enthusiasts because of its high protein and low lipid content [2]. It is worth mentioning that studies have shown that the composition of amino acids in beef is very close to the amino acids needed by the human body, which has a significant effect on enhancing human immunity. In recent years, with improved living standards and an in-depth understanding of nutrition knowledge, people’s demand for beef has increased rapidly [3]. Among the various forms of beef prepared for consumers, chilled beef refers to the slaughter of beef and direct processing at ultra-low temperature to maintain the internal temperature of the beef at 4 °C. It is discharged with a high standard of acid treatment in the subsequent processing, circulation, and sales process, which are always carried out at 4 °C [4]. Due to its appropriate storage temperature conditions, chilled beef maintains the original organizational structure and nutritional composition of beef, which makes the chilled beef rich in nutritional value, good quality, and conducive to human digestion and absorption. However, chilled beef is highly perishable, making it an ideal vehicle for rapid microbial growth and considerable economic losses. Therefore, it is of utmost importance for the beef industry to find novel and effective preservation materials to maintain the quality of chilled beef and increase its shelf life [5].

In recent years, due to the limited disposal methods of plastic packaging waste, its impact on the environment has gained much attention. Non-biodegradable plastic packaging materials increase the burden of environmental pollution and consume many natural resources. Therefore, the emergence of biodegradable packaging materials better addresses the food safety problems. Based on the natural biological polymer, the perfect replacement of synthetic plastic packaging materials with biodegradable packaging materials has become a focus of new research to prepare novel biodegradable polymer materials [6,7].

The bio-nanocomposite film is a biodegradable packaging material based on natural biological polymer, a new polymer used in the food packaging market. Bio-nano polymer has excellent mechanical and barrier properties and is biodegradable at the end of its life. The application of nanotechnology has effectively overcome the inherent limitations of natural biopolymer packaging materials, such as low mechanical strength and low water resistance. Today, natural polymer-based composites have great potential to improve the quality and safety of packaged foods by improving their antimicrobial activity and barrier properties [8].

Due to their therapeutic effects, in recent years, plant EOs have been widely used in food packaging and preservation to ensure food safety and quality [9]. In one recent study, clove EO loaded nanoemulsion and pickering emulsion activated pullulan-gelatin based edible film exhibited excellent mechanical properties, water barrier properties, and appreciable antioxidant activities [10]. Chitosan/montmorillonite bio-nanocomposites incorporated with rosemary and ginger EOs could reduce lipid oxidation and microbiological contamination and extend the preservation effect on the fresh poultry meat [11]. Souza et al [12,13] characterized chitosan/montmorillonite films incorporated with ginger EOs to improve the shelf life of fresh poultry meat. Souza et al further evaluated the activity of these bio-nanocomposites incorporated with rosemary EOs in fresh poultry meat [14]. Hamedi et al [15] reported that sodium alginate polymer incorporated with EOs of *Ziziphora persica* could inhibit (*p* < 0.05) the growth and reproduction of *Escherichia coli* (*E. coli*), *Pseudomonas aeruginosa*, and *Listeria monocytogenes* and improve the quality of the chicken during refrigeration. Perdones et al [16] combined chitosan with lemon EOs to form a coating for strawberry freshness preservation and found that the chitosan coating could promote ester formation in a short time, and the conjugate could transfer terpenoid volatiles to the fruit, making the original fruit flavor more obvious. Moradi et al [17] reported the inhibitory effect of edible zein film impregnated with *Zataria multiflora* Boiss. EOs and monolaurin on *L. monocytogenes* and *E. coli* in beef. More and more studies report on the preparation and application of biodegradable nanocomposites, which are expected to replace or reduce existing petrochemical packaging materials and become a research hotspot for functional bio-composite packaging materials.

In the previous study, we reported the antibacterial activity and mechanism of GEO against *E. coli* and *Staphylococcus aureus* [18]. In present research, the polysaccharide chitosan as the polymer matrix was combined with nano-clay montmorillonite and natural active substance GEO to prepare GEO, chitosan, and montmorillonite novel bio-nanocomposite films. The composite films were tested for antioxidant and antibacterial properties, and their sustained-release performance, safety, and stability attributes were also evaluated. Our results demonstrated that the resulting bio-nano films not only overcame the problem of auto-oxidation of GEO but they could also improve the properties of chitosan film. Therefore, the prepared biological nanocomposites could be used as promising materials to replace commercial and non-degradable plastic films.

## 2. Results and Discussion

### 2.1. FT-IR Analysis of the Bio-Nanocomposites Film

FT-IR spectra of composite films prepared by different formulations are shown in Figure 1. FT-IR is a commonly used technique for identifying chemical components and their internal molecular functional group interactions as valid. The infrared spectra of the original chitosan film were also observed.

The characteristic absorption peaks were observed at 3325 cm^−1^ (axial stretch-OH), 3273 cm^−1^ (asymmetrically stretched -NH group), 2882–2934 cm^−1^ (C-H bond of methyl), 1638 cm^−1^ (amide group), at 1403–1550 cm^−1^ (amide group), bone vibrations at 1342 cm^−1^ (involving amide C-N bond stretching), 1342–1403 cm^−1^ (-CH2 folded), 874–1143 cm^−1^ (bone vibration involved in C-O bond stretching), and at 1134 cm^−1^ (C-O-C bond asymmetrical stretching) [19,20]. Compared with the original chitosan film, adding GEO and MMT showed a minor difference in the spectra, which may be due to the small volume and small content of MMT and GEO. In Figure 1, the signal at 1550 cm^−1^ almost disappeared when the percentage of GEO increased, which might be due to the amide group reaction with GEO to form new functional groups. The characteristic peaks of chitosan were found in all the samples. However, some changes in the intensity of absorption peaks were also observed due to the overlapping chemical bonds of absorption peaks, thus, indicating a strong interaction that was consistent with Souza et al [13].

### 2.2. Thickness and Mechanical Properties of Films

The thickness and mechanical properties of different fresh-keeping films are shown in Table 1. The results showed that the addition of MMT did not significantly change the thickness of the films (*p* > 0.05). In addition, the thickness of the films with MMT was slightly smaller than that without MMT. The reason may be due to the penetration of the chitosan chain, which causes a robust force between the polymer and MMT, forming a compact structure [21]. On the other hand, with the increase of GEO concentration, the thickness of the film was significantly increased (*p* < 0.05). This may be due to the interaction between chitosan and GEO, which reduces the arrangement of polymer chains and reduces the compression required to form the network, thus, increasing the thickness of the film [22,23].

The mechanical property of the film is one of the most important basic properties of the packaging film [24], which is used to maintain the integrity of the product and minimize the damage of the product caused by external forces. The results showed that in the films without natural extract (group of chitosan and chitosan + MMT), the nanoclay significantly increased the tensile strength and elongation at break of the films (*p* < 0.05). This is mainly due to the strong interaction between chitosan and montmorillonite, making the montmorillonite uniformly dispersed in the chitosan system (exfoliation and intercalation phenomena) [25]. In addition, the infiltration of GEO reduced the tensile resistance of the film while increasing the elongation at break. This might be due to the combination of lipids and polymers, resulting in a weak interaction between chitosan and GEO, which replaced some strong polar chemical bonds between chitosan molecules and led to the formation of heterogeneous and discontinuous structures in the film matrix, thus, affecting the mechanical properties of the film [26].

### 2.3. Water Vapor Permeability of Films

In the development of new food packaging materials, it is imperative to study the barrier properties of materials. The barrier properties play a physical protective role so that the shelf life of packaged food is prolonged. The WVP of different fresh-keeping films is shown in Table 2. The results showed that the addition of MMT and GEO did not significantly change the WVP of films (*p* > 0.05), except that the ratio of chitosan + MMT + 0.5% GEO group was significantly higher than that in other groups (*p* < 0.05). At the same time, it was found that water vapor transmittance increased with the increase of the GEO proportion, indicating that the water vapor barrier was decreased. Ideally, with the incorporation of hydrophobic substances (EOs), the film should reduce the water vapor permeability because the water vapor transfer process depends on the ratio of hydrophilic/hydrophobic components. Previous studies attributed the increase in water vapor transmittance to the possible interactions between hydrophobic substances and chitosan components that make the polymer matrix more open to the transport of water molecules [27]. In general, several films showed appreciable barrier properties with a great application prospect in food preservation.

### 2.4. Migration of Phenolic Components of Films

The release of phenolic compounds to 95% ethanol followed a migration pattern of exponential increase to the maximum and reached equilibrium after 24–48 h of contact with the mimics. The results showed that with the increase of the EO concentration, the total phenol content in the simulation was significantly increased (*p* < 0.05) (Figure 2). At the same time, the addition of MMT slightly reduced the effective components of the diffusion process, which was consistent with the previous studies [12,28]. This phenomenon can be attributed to the interactions between hydroxy and amino groups of chitosan matrix, which lead to a special arrangement and GEO infiltration and enhance the chitosan and the hydrogen bonding interaction between MMT. As a result, the network structure formed between MMT and chitosan encapsulated the phenolic compounds in GEO and restricted their diffusion. The most significant advantage of active packaging in the food system is the extension of protection time through the gradual release of bioactive compounds. Therefore, the samples with the highest concentrations of GEO and MMT obtained from this in vitro migration test can be used as the most promising films.

### 2.5. Physicochemical Index of Chilled Beef Preserved by Bio-Composite Films

The changes of pH, hue angle value, and moisture value of chilled beef in different times of film preservation are shown in Table 3. In terms of pH value, the unpackaged samples and the samples packed with the original chitosan film had a significantly higher pH value after 15 d of storage (*p* < 0.05). There was no significant difference in pH values (*p* > 0.05) for all the other samples encased in the bio-composites, although there was a slight increase compared to the d 1. At the same time, there was no significant difference between the groups containing GEO due to the increase of EO concentration (*p* > 0.05). On d 15, the pH value of the untreated group was significantly higher than that of other groups (*p* < 0.05), and there was no significant difference in the value of the original chitosan group and chitosan + MMT group (*p* > 0.05), but it was still higher than that of other groups. Previous reports [29,30] indicated that for foods rich in protein and free amino acids stored under aerobic conditions, the increase in pH might be due to volatile essential components from microbial growth on the food surface, i.e., their endogenous proteolysis leads to the production of alkaline components. Therefore, the antibacterial ability of the film may be the reason for the low pH value of the sample.

In terms of the hue angle values of each group, the initial hue value was about 51°, showing a red tint. After 15 d of storage, the value of the unpacked group increased to about 70°, while the value of the treated group was about 55°. The changing of the hue angle from small to large means that the color of the fresh meat changed from red to more yellow and green. There was no significant difference in the hue angle value of the composite film treatment group (*p* > 0.05), which effectively maintained fresh beef color. Previous studies [29,31] attributed the increase in tonal angle values to lipid oxidation in meat and the degradation of heme molecules to form myoglobin. Therefore, the composite film may effectively delay the modification process. These results demonstrate the potential of this composite material for the preservation of fresh meat.

In terms of moisture value, the water content of meat can also be an important reference index for consumers. Compared with the untreated group, the moisture value of the composite film treatment group was maintained or decreased over time, especially the film containing GEO. It has been reported that chitosan itself is a hydrophilic polysaccharide that can absorb water in meat, and it exhibits an appreciable air permeability at the same time. In the previous studies [13], poultry meat with rosemary EO and MMT in composite film packaging showed similar results, possibly due to the chitosan’s water absorption and air permeability, which confirmed the present results.

### 2.6. Lipid Oxidation Value of Chilled Beef Preserved by Bio-Composite Films

The changes in lipid oxidation value of chilled beef preserved by bio-composite films at different times are shown in Table 4. TBARS value is usually a critical index to evaluate the degree of lipid oxidation in meat and a measurement index of the MDA content, a product of fatty acid oxidation and degradation in meat [29]. It has been reported that the threshold for consumer odor perception corresponds to a sample with a TBARS value of 0.5 mg MDA/kg [30]. In terms of fat oxidation value, the TBARS value of the initial sample was 0.24 mg MDA/kg, and the TBARS value of the untreated group reached 2.05 mg MDA/kg at 15 d, indicating that the beef had begun to rot. The TBARS value of the treated group was significantly lower than that of the untreated group (*p* < 0.05), except that the original chitosan film and the chitosan + MMT group exceeded the threshold value at 15 d.

Moreover, all the other groups had lower than 0.5 mg MDA/kg, indicating that several films had a specific preservation effect on chilled beef. At the same time, the TBARS value of the treated group with MMT was higher than that without MMT. It was previously reported [20,24] that the addition of MMT decreased the activity of the film, which may be due to the enhanced molecular interaction between components of the bio-materials and interfered with the migration of active compounds, thus, reducing their anticorrosion ability [32].

### 2.7. Total Mesophilic Aerobic Bacteria of Chilled Beef Preserved by Bio-Composite Films

The changes of TMAB in chilled beef preserved by the composite film at different time are shown in Table 4. A previous study reported that the maximum allowable TMAB value for meat was set at 10^7^ CFU/g before it was unsuitable for consumption [14]. Along with lipid oxidation, microbial growth is one of the most destructive processes in food. In terms of the TMAB, the initial TMAB value of the sample was 4.6, and then it showed a significant increasing trend with the increase of time (*p* < 0.05). Among them, the TMAB value of the untreated group on d 10 was 9.6, and the experimental group also showed different degrees of TMAB increase, indicating the spoilage of fresh beef. At the same time, with the increase in the concentration of GEO, the TMAB value decreased to some extent, but there was no significant difference (*p* > 0.05). In addition, like the lipid oxidation value, the TMAB value of the MMT added to the group was higher than that of the non-added MMT group, indicating that the addition of MMT reduced the anti-corrosion effect of the bio-nanocomposite, leading to a higher TMAB value.

To summarize, the bio-materials demonstrated effective preservation effects, and the nanocomposite films were identified as potential preservative films, effectively extending the shelf life of meat. The bio-nanocomposite films containing different proportions of ginger essential oil (GEO) might be convenient and effective technology for beef preservation.

## 3. Materials and Methods

### 3.1. Materials and Reagents

As described in our previous study, food-grade ginger (*Zingiber officinale* Roscoe) essential oil was obtained [18]. The fresh beef tenderloin was purchased from the local market in Hefei, Anhui province. Montmorillonite and chitosan were procured from Shanghai Macklin Biochemical Co., Ltd., China. All the chemical reagents used in this study were of analytical grade.

### 3.2. Preparation of Bio-Nanocomposite Films

The bio-nanocomposite films were prepared according to the previously described methods [11,14]. Briefly, 1.5 g of chitosan was added into a beaker containing 100 mL of 1% glacial acetic acid solution. The beaker was placed on a magnetic stirrer allowed to stir overnight. When the solution was completely dissolved, 0.45 g glycerin was added as a plasticizer, and then 2.5% MMT (*w*/*w* chitosan) was added. The solution was stirred and homogenized three times at a speed of 15000 rpm for 5 min each, using a digital display high-speed stirring and dispersing machine (MA60, Shanghai Ouhe Machinery Equipment Co., Ltd., Shanghai, China), followed by an ultrasonic water bath for 10 min. Before the final stirring, the various volume concentrations of GEO (0.1%, 0.3%, 0.5% *v*/*v* mixture) were added to the mixture system, followed by the addition of Tween 80 (0.2% *v*/*v* GEO). The resulting mixed system was formed into a bio-nanocomposite film in a polymer mold (15 cm × 15 cm) and dried at room temperature for 72 h (50% relative humidity). The eight kinds of films prepared by the above method were preserved at 25 °C after peeling.

### 3.3. Characterization of Bio-Nanocomposite Films

#### 3.3.1. Fourier Transform Infrared Spectroscopy (FT-IR)

The films were pretreated, and the FT-IR spectra of the samples were determined by the KBr compression method. The KBr slices were taken as blank control to collect the baseline. The composite films were mixed with KBr and pressed into thin sheets and then examined by FTIR spectrometer (Nicolet iS50, Unico Instrument Co., Ltd. America) at the scanning wavelength range of 4000–400 cm^−1^ [33].

#### 3.3.2. Thickness and Mechanical Properties of Films

The thickness of films (length and width of 60 mm and 20 mm, respectively) was measured by a digital display vernier caliper [34]. According to GB/T 1040.3 standard [35], tensile strength and elongation at the break of composite films were measured using a texturing apparatus and tensile probe. The test conditions were as follows: clamping distance was 30 mm, and the test speed was 6 cm/min. The tensile strength and elongation break were calculated according to the following formula:(1)$$\mathrm{Ts}=\frac{\;F\;}{\;S\;}$$
(2)$$\mathrm{EB}=\frac{{\;L}_{1}{\;-\;L}_{0}}{{\;L}_{0}}\text{×}100\%\;$$
where F—Maximum tension at fracture, N; S—Cross-sectional area, mm^2^; L_0_—Original length, mm; L_1_—length at fracture, mm; Ts—tensile strength, MPa; EB—elongation at break (%).

#### 3.3.3. Determination of Water Vapor Permeability of Bio-Nanocomposite Films

0.2 g of the film was sealed in a glass bottle. The average diameter of the film was 2 cm, and the height was 4.5 cm, which was slightly larger than the diameter of the glass bottle. The glass bottle contained 3 g anhydrous calcium chloride with a relative humidity of 0% until constant. Then, the dry glass bottle was placed in a 50 mL beaker filled with saturated potassium sulfate at 98% saturated humidity and 25 °C and weighed every 24 h. The weight change was recorded as a function of time, and the slope was calculated by linear regression (weight and time). The water vapor transport rate was defined as the linear slope of the curve (g/h) divided by the transport area (m^2^), and the water vapor transmittance was calculated according to the previous study [36].
(3)$$\mathrm{WVP}=\frac{\mathrm{WVTR}\;\times \;d\;}{{\;P\;\times \;(\;R}_{1}{\;-\;R}_{2\;})\;}$$
where WVTR—water vapor transport rate; d—film thickness, m; P—saturated vapor pressure of water at 25 °C, Pa; R_1_—relative humidity of glass bottle (%); R_2_—relative humidity (%); WVP—water vapor permeability (g/m·H·Pa).

#### 3.3.4. Migration Determination of Active Components in Films

The release amount of the active compounds in the composite film was determined by the Folin phenol method, which was determined by the content of the phenolic compounds that migrated into 95% ethanol solution (fatty food simulation) at 37 °C. Briefly, 2 mL sample solution was mixed with 3 mL ultrapure water and 0.5 mL Folinol reagent, placed in the dark for 10 min, followed by the addition of 1 mL of 5% (W/V) sodium carbonate solution, mixing, placing in the dark place for about 60 min, and then an absorbance measurement at 760 nm wavelength. The equivalent concentration of gallic acid was used to express the total polyphenol content released [12,37].

### 3.4. Characterization of Chilled Beef

#### 3.4.1. Fresh Beef Pretreatment

First, alcohol (95%) was used to disinfect the cutting tool and cutting board, and the composite film and Petri dishes were sterilized under UV exposure in the ultra-clean table. The fresh beef tenderloin was cut into equal-sized pieces (20 g) on the ultra-clean table and wrapped with composite film and placed in the Petri dishes after sterilization, corresponding to the experimental group, control group, and blank group, respectively. The meat in the experimental group was treated with the composite film containing GEO, the control group was treated with the composite film without GEO, and the blank group was without any treatment. After treatment, the samples were stored in a refrigerator at 4 °C for 15 days. Samples were randomly sampled at 0 d (3 h after treatment), 3 d, 7 d, 10 d, and 15 d after treatment. Three samples were taken from each treatment group, and the following indexes were measured sequentially.

#### 3.4.2. Physicochemical Index

According to GB 5009.237 standard [38], The 5 g of beef was homogenized with 50 mL deionized water in a digital display high-speed stirring and dispersing machine for 15 min. Calibrated pH meter was used for the pH measurement. The moisture content was determined after oven drying using the measurement method of GB 5009.3 standard [39]. Hue angle [14] was measured; after the meat sample was taken out of the refrigerator, the water on the sample surface was wiped with filter paper, and then the color difference meter was used to measure the point on the sample surface, and the L*, a*, and b* values measured by the color difference meter were recorded. Each sample was tested three times to calculate the hue angle:(4)$$\mathrm{Hue}\;\mathrm{angle}=\tan(\frac{b^{*}}{a^{*}}{)}^{-1}$$

#### 3.4.3. Determination of Lipid Oxidation Value 

For this, the thiobarbituric acid reactive substances (TBARS) method was slightly modified according to the method reported previously [40]. The 10 g meat samples were mixed with 20 mL trichloroacetic acid and stirred for 1 h to extract malondialdehyde (MDA). Five milliliters of the filtrated supernatant were added with 0.02 mol/L thiobarbituric acid and placed in a water bath at 95 °C and heated for 30 min. After the water was cooled, the absorbance of the sample was measured at 530 nm with a spectrophotometer (UV-2100, Unico (Shanghai) Instrument Co., Ltd., Shanhai, China). Results were quantified as known MDA concentration, expressed in units per mg MDA/kg.

#### 3.4.4. Determination of Total Mesophilic Aerobic Bacteria (TMAB)

According to GB 4789.2 Standard [41], TMAB was used for microbiological evaluation of the quality of packaged meat. TMAB counts were performed in PCA after incubation at 30 °C for 72 h.

### 3.5. Statistical Analysis

SPSS Statistics 20.0 was used for Duncan’s test for data analysis at the significance level of *p* < 0.05. All experiments were performed in triplicate, and data were presented as mean ± standard deviation.

## 4. Conclusions

In this study, GEO, chitosan, and MMT were used as materials to prepare several kinds of edible bio-composite films. The film showed good mechanical properties and barrier properties, good slow-release properties, and slow migration of active components. Such films can be considered as potential environmentally friendly films for active packaging materials. The composite film preservation showed that the composite film could effectively reduce the rise of pH value, hue angle value, and moisture value of chilled beef with time, as well as delay the fat oxidation and the growth of surface microorganisms on chilled beef. Present research demonstrated that the bio-nanocomposite films containing different proportions of ginger essential oil (GEO) might be convenient and effective technology for beef preservation.

## Figures and Tables

**Figure 1 antibiotics-10-00796-f001:**
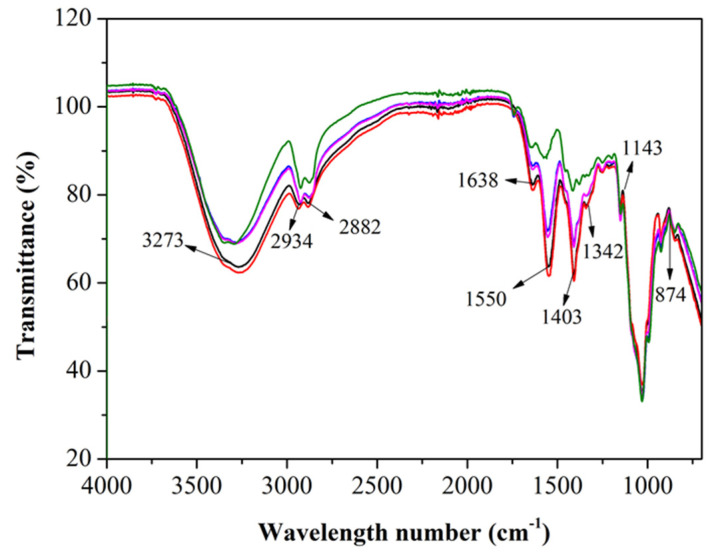
Fourier transform infrared spectra of (Black) Ch, (Orange) Ch + MMT, (Blue) Ch + MMT + 0.1% GEO, (Pink) Ch + MMT + 0.3% GEO, (Green) Ch + MMT + 0.5% GEO. The spectra data were obtained in the range of 4000 to 400 cm^−1^ at the resolution of 4 cm^−1^ and with 56 scans per spectrum.

**Figure 2 antibiotics-10-00796-f002:**
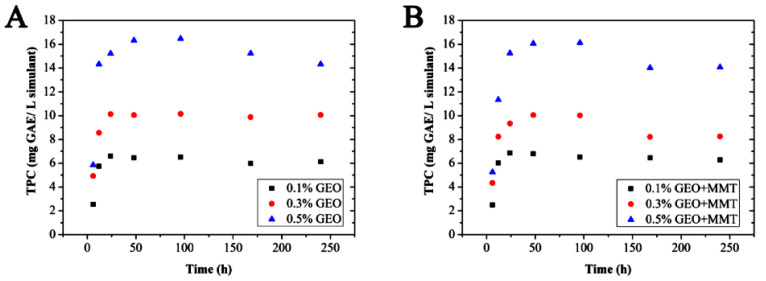
Migration of active components in different kinds of fresh-keeping films. (**A**), the transferring value of total phenol content in the composite film containing ginger essential oil. (**B**), the Transferring value of total phenol content in the composite film rich in montmorillonite and ginger essential oil. The migration of active components was determined using the Folin phenol method, which was determined by the content of the phenolic compounds that migrated into 95% ethanol solution at 37 °C.

**Table 1 antibiotics-10-00796-t001:** Thickness and mechanical properties of different fresh-keeping films.

Types of Film	Thickness (µm)	Ts (MPa)	EB (%)
Ch	44 ± 1.2 ^a^	48.6 ± 1.5 ^A^	19.7% ± 1.8 ^a^
Ch + MMT	43 ± 0.8 ^a^	62.4 ± 3.1 ^B^	32.2% ± 4.2 ^b^
Ch + 0.1% GEO	56 ± 1.1 ^b^	46.5 ± 4.2 ^A^	24.5% ± 3.7 ^a^
Ch + 0.3% GEO	63 ± 2.1 ^bc^	33.4 ± 2.3 ^C^	22.1% ± 4.5 ^a^
Ch + 0.5% GEO	73 ± 2.3 ^c^	30.2 ± 1.1 ^C^	30.1% ± 2.5 ^b^
Ch + MMT + 0.1% GEO	53 ± 0.4 ^b^	42.4 ± 2.7 ^A^	32.1% ± 2.3 ^b^
Ch + MMT + 0.3% GEO	61 ± 2.1 ^bc^	30.6 ± 1.8 ^C^	30.3% ± 3.7 ^b^
Ch + MMT + 0.5% GEO	68 ± 2.5 ^c^	28.6 ± 3.1 ^C^	34.5% ± 6.2 ^b^

Notes: Ts—Tensile strength; EB—the percentage of breaking elongation; ^A–C^, different letters represent the comparison between groups on the same line, *p* < 0.05; ^a–c^, different letters for comparison with the same column group, *p* < 0.05.

**Table 2 antibiotics-10-00796-t002:** Water vapor permeability of different kinds of fresh-keeping films.

Types of Film	WVP (10^−7^ g/m·h·Pa)
Ch	9.18 ± 0.23 ^a^
Ch + MMT	9.82 ± 0.32 ^ab^
Ch + 0.1% GEO	10.32 ± 0.31 ^abc^
Ch + MMT + 0.1% GEO	10.97±0.33 ^abc^
Ch + 0.3% GEO	10.42 ± 0.46 ^abc^
Ch + MMT + 0.3% GEO	9.65 ± 0.49 ^ab^
Ch + 0.5% GEO	11.88 ± 0.74 ^abc^
Ch + MMT + 0.5% GEO	15.79 ± 0.84 ^c^

Notes: WVP—Water vapor permeability; ^a–c^, different letters for comparison with the same column group, *p* < 0.05.

**Table 3 antibiotics-10-00796-t003:** Changes of color value, pH value and water activity value of chilled beef after membrane preservation for different times.

Parameters	Days	Unwrapped	Ch	Ch + 0.1% G	Ch + 0.3% G	Ch + 0.5% G	Ch+MMT	Ch + MMT + 0.1% G	Ch + MMT + 0.3% G	Ch + MMT + 0.5% G
Hue angle (°)	0	50.6 ± 1.8 ^a,A^	49.7 ± 1.2 ^a,A^	51.4 ± 1.8 ^a,A^	51.1 ± 1.2 ^a,A^	50.8 ± 1.5 ^a,A^	50.4 ± 1.3 ^a,A^	50.1 ± 1.4 ^a,A^	51.3 ± 2.1 ^a,A^	52.3 ± 2.1 ^a,A^
	3	51.9 ± 1.6 ^a,A^	50.2 ± 1.1 ^a,A^	54.2 ± 2.0 ^b,B^	55.6 ± 0.9 ^b,BC^	58.7 ± 2.0 ^b,C^	46.1 ± 1.1 ^b,D^	47.3 ± 1.6 ^b,D^	57.7 ± 0.8 ^b,C^	58.9 ± 1.9 ^b,C^
	7	58.2 ± 2.1 ^b,A^	52.4 ± 1.6 ^b,B^	55.2 ± 1.7 ^b,AB^	56.1 ± 1.1 ^b,AB^	54.5 ± 1.9 ^ab,AB^	53.2 ± 0.9 ^c,AB^	58.2 ± 0.8 ^c,A^	58.3 ± 2.1 ^b,A^	55.3 ± 2.3 ^ab,AB^
	10	61.5 ± 2.6 ^c,A^	50.6 ± 1.4 ^a,B^	54.5 ± 2.2 ^b,C^	52.5 ± 1.5 ^a,BC^	53.6 ± 1.9 ^ab,BC^	53.9 ± 0.6 ^c,BC^	52.7 ± 1.2 ^a,BC^	49.8 ± 0.8 ^a,B^	50.5 ± 1.3 ^a,B^
	15	70.1 ± 1.1 ^d,A^	55.6 ± 1.5 ^c,B^	55.5 ± 1.5 ^b,B^	54.9 ± 0.8 ^b,B^	54.3 ± 1.3 ^ab,B^	54.3 ± 0.7 ^c,B^	53.3 ± 1.3 ^ac,B^	52.8 ± 1.1 ^a,B^	53.7 ± 1.4 ^a,B^
pH	0	5.87 ± 0.8 ^a,A^	5.87 ± 0.4 ^a,A^	5.91 ± 0.2 ^a,A^	5.86 ± 0.8 ^a,A^	5.98 ± 0.4 ^a,A^	5.92 ± 0.3 ^a,A^	5.87 ± 0.0 ^a,A^	5.85 ± 0.2 ^a,A^	5.93 ± 0.4 ^a,A^
	3	6.16 ± 0.5 ^ab,A^	6.02 ± 0.5 ^a,A^	5.95 ± 0.7 ^a,A^	5.96 ± 0.5 ^a,A^	6.04 ± 0.6 ^a,A^	6.01 ± 0.4 ^a,A^	5.93 ± 0.3 ^a,A^	5.92 ± 0.1 ^a,A^	5.96 ± 0.2 ^a,A^
	7	6.72 ± 0.3 ^b,A^	6.20 ± 0.2 ^a,B^	6.12 ± 0.3 ^a,B^	6.05 ± 0.1 ^a,B^	5.99 ± 0.1 ^a,B^	6.32 ± 0.2 ^b,C^	6.22 ± 0.0 ^b,B^	6.07 ± 0.4 ^a,B^	6.04 ± 0.1 ^a,B^
	10	8.38 ± 0.2 ^c,A^	7.30 ± 0.1 ^b,B^	6.87 ± 0.2 ^b,C^	7.05 ± 0.4 ^b,BC^	6.63 ± 0.1 ^b,C^	7.22 ± 0.4 ^c,B^	7.04 ± 0.1 ^c,BC^	6.96 ± 0.4 ^b,BC^	6.49 ± 0.1 ^b,C^
	15	8.33 ± 0.1 ^c,A^	7.33 ± 0.3 ^b,A^	6.90 ± 0.5 ^b,A^	6.87 ± 0.3 ^b,A^	6.75 ± 0.3 ^b,A^	7.30 ± 0.1 ^c,A^	7.15 ± 0.4 ^c,A^	7.04 ± 0.3 ^b,A^	6.74 ± 0.3 ^c,A^
Moisture(%)	0	74.6 ± 1.1 ^a,A^	74.6 ± 1.4 ^a,A^	74.5 ± 0.9 ^a,A^	74.6 ± 0.7 ^a,A^	74.3 ± 1.1 ^a,A^	74.5 ± 0.5 ^a,A^	74.6 ± 0.2 ^a,A^	74.5 ± 1.1 ^a,A^	74.5 ± 0.7 ^a,A^
	3	73.6 ± 0.8 ^b,A^	72.6 ± 0.5 ^b,B^	72.5 ± 1.2 ^b,B^	72.4 ± 0.4 ^b,B^	71.2 ± 0.6 ^b,C^	72.3 ± 0.6 ^b,B^	71.3 ± 0.4 ^b,C^	69.7 ± 1.0 ^b,D^	69.8 ± 0.9 ^b,D^
	7	75.1 ± 1.3 ^a,A^	72.7 ± 1.1 ^b,B^	72.8 ± 0.8 ^b,B^	71.9 ± 0.2 ^b,B^	72.0 ± 0.4 ^b,B^	72.1 ± 0.3 ^b,B^	70.8 ± 0.3 ^bc,C^	69.4 ± 0.6 ^b,D^	69.6 ± 1.2 ^b,D^
	10	76.8 ± 1.6 ^c,A^	72.3 ± 0.8 ^b,B^	72.6 ± 0.7 ^b,B^	72.1 ± 0.9 ^b,B^	71.8 ± 1.4 ^b,B^	71.9 ± 0.3 ^b,B^	70.3 ± 0.5 ^c,C^	69.1 ± 0.8 ^b,D^	69.2 ± 1.1 ^b,D^
	15	76.9 ± 0.7 ^c,A^	72.0 ± 0.6 ^b,B^	72.4 ± 1.1 ^b,B^	71.8 ± 1.2 ^b,B^	71.6 ± 0.9 ^b,B^	71.7 ± 1.1 ^b,B^	70.4 ± 0.8 ^c,C^	68.9 ± 1.2 ^b,D^	69.3 ± 0.7 ^b,D^

Note: ^A–D^, different letters represent the comparison between groups on the same line, *p* < 0.05; ^a–d^, different letters for comparison with the same column group, *p* < 0.05.

**Table 4 antibiotics-10-00796-t004:** Changes of fat oxidation value and total microorganism content in chilled beef after film storage for different times.

Parameters	Days	Unwrapped	Ch	Ch + 0.1% G	Ch + 0.3% G	Ch + 0.5% G	Ch + MMT	Ch + MMT + 0.1% G	Ch + MMT + 0.3% G	Ch + MMT + 0.5% G
TBARS (mg MDA/kg meat)	0	0.24 ± 0.02 ^a,A^	0.24 ± 0.02 ^a,A^	0.24 ± 0.02 ^a,A^	0.24 ± 0.02 ^a,A^	0.24 ± 0.02 ^a,A^	0.24 ± 0.02 ^a,A^	0.24 ± 0.02 ^a,A^	0.24 ± 0.02 ^a,A^	0.24 ± 0.02 ^a,A^
	3	0.28 ± 0.01 ^a,A^	0.27 ± 0.01 ^a,A^	0.31 ± 0.01 ^ab,A^	0.30 ± 0.02 ^a,A^	0.26 ± 0.02 ^a,A^	0.28 ± 0.03 ^a,A^	0.27 ± 0.03 ^a,A^	0.27 ± 0.03 ^a,A^	0.26 ± 0.01 ^a,A^
	7	1.03 ± 0.08 ^b,A^	0.42 ± 0.02 ^ab,B^	0.36 ± 0.07 ^b,B^	0.32 ± 0.03 ^a,B^	0.31 ± 0.01 ^a,B^	0.48 ± 0.01 ^b,B^	0.29 ± 0.06 ^a,B^	0.38 ± 0.03 ^b,B^	0.35 ± 0.04 ^b,B^
	10	1.62 ± 0.06 ^c,A^	0.46 ± 0.01 ^ab,B^	0.38 ± 0.05 ^b,B^	0.33 ± 0.04 ^a,B^	0.31 ± 0.01 ^a,B^	0.52 ± 0.06 ^b,B^	0.32 ± 0.01 ^a,B^	0.36 ± 0.02 ^b,B^	0.34 ± 0.01 ^b,B^
	15	2.05 ± 0.03 ^d,A^	0.51 ± 0.05 ^b,BC^	0.41 ± 0.06 ^b,B^	0.32 ± 0.05 ^a,C^	0.33 ± 0.06 ^a,C^	0.69 ± 0.05 ^c,B^	0.48 ± 0.06 ^b,BC^	0.43 ± 0.05 ^b,B^	0.38 ± 0.06 ^b,B^
TMAB (Log10CFU/g meat)	0	4.6 ± 0.1 ^a,A^	4.6 ± 0.1 ^a,A^	4.6 ± 0.1 ^a,A^	4.6 ± 0.1 ^a,A^	4.6 ± 0.1 ^a,A^	4.6 ± 0.1 ^a,A^	4.6 ± 0.1 ^a,A^	4.6 ± 0.1 ^a,A^	4.6 ± 0.1 ^a,A^
	3	6.2 ± 0.2 ^b,A^	6.0 ± 0.2 ^b,A^	6.0 ± 0.2 ^b,A^	6.0 ± 0.2 ^b,A^	5.9 ± 0.1 ^b,A^	6.0 ± 0.2 ^b,A^	5.5 ± 0.1 ^b,B^	5.9 ± 0.1^b,A^	5.9 ± 0.2 ^b,A^
	7	9.1 ± 0.4 ^c,A^	8.4 ± 0.3 ^c,B^	7.9 ± 0.3 ^c,BC^	7.3 ± 0.2 ^c,C^	7.1 ± 0.5 ^c,C^	8.6 ± 0.4 ^c,B^	8.2 ± 0.6 ^c,B^	8.1 ± 0.2 ^c,B^	7.8 ± 0.2 ^c,BC^
	10	9.6 ± 0.3 ^c,A^	8.7 ± 0.4 ^c,B^	8.0 ± 0.3 ^c,B^	7.6 ± 0.1 ^c,BC^	7.3 ± 0.1 ^c,C^	8.9 ± 0.3 ^c,B^	8.4 ± 0.7 ^c,B^	8.3 ± 0.1 ^c,B^	8.0 ± 0.2 ^c,B^

Notes: ^A–C^, different letters represent the comparison between groups on the same line, *p* < 0.05; ^a–d^, different letters for comparison with the same column group, *p* < 0.05.

## Data Availability

Present paper no this concern.

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
