# Peer review of "Preparation and Characterization of Bio-Nanocomposites Film of Chitosan and Montmorillonite Incorporated with Ginger Essential Oil and Its Application in Chilled Beef Preservation"

_antibiotics, 2021, doi:10.3390/antibiotics10070796_

Round 1

Reviewer 1 Report

Reviewer report

Zhang et al., prepared and characterized bio-nano composite films containing different proportions of ginger essential oil (GEO), Chitosan (Ch), and Montmorillonite (MMT) and studied the preservation effect of bio-nano composite films on chilled beef. It seems to me an important manuscript and can be considered for publication in this Journal. However, following suggestions are recommended:

  1. There are notable mistakes throughout the manuscript in choosing English word and typing etc. Authors are recommended to take help from an native English speaker or English expert if possible to improve the quality of the manuscript to a notable extent.
  2. Author need to revise the whole abstract carefully to make it stand alone especially use of improper punctuation mark.
  • In the last paragraph of the introduction, the Author needs to clearly state the novelty of this paper together with future prospects of this study.
  1. Authors need to follow the journal format fully in the case of the Reference list. For example, Journal abbreviations, heading, and subheadings etc.
  2. In the result and discussion section, the author needs to pay more attention and validate their findings with recent previous results and compare if possible.
  3. All Figure caption needs to be updated including detailed experimental conditions as precisely as possible. The presentation of the Figure needs to be improved.
  • The conclusion part needs to be revised and English language usage mistakes must be removed.

Author Response

Zhang et al., prepared and characterized bio-nano composite films containing different proportions of ginger essential oil (GEO), Chitosan (Ch), and Montmorillonite (MMT) and studied the preservation effect of bio-nano composite films on chilled beef. It seems to me an important manuscript and can be considered for publication in this Journal. However, following suggestions are recommended:

Comment 1: There are notable mistakes throughout the manuscript in choosing English word and typing etc. Authors are recommended to take help from an native English speaker or English expert if possible to improve the quality of the manuscript to a notable extent.

Response 1: Thank you very much. We invited Dr. Kannan RR Rengasamy from University of Limpopo to revise the grammar of the text.

Comment 2: Author need to revise the whole abstract carefully to make it stand alone especially use of improper punctuation mark.

Response 2: Thank you very much. Abstract in manuscript was revised as follow:

In this study, bio-nano composite films containing different proportions of ginger essential oil (GEO), Chitosan (Ch), and Montmorillonite (MMT) were prepared and characterized, and the antibacterial effect of bio-nano composite films on chilled beef was evaluated. Fourier transform infrared analysis showed a series of intense interactions among the components of the bio-nano composite films. The infiltration of GEO increased the thickness of the film, reduced the tensile strength of the film, and increased the percentage of breaking elongation and the water vapor permeability. The migration of phenols in the films began to increase exponentially and reached equilibrium at about 48 h. The bio-nano composite films (Ch +0.5% GEO group, and Ch +MMT+0.5% GEO group) effectively delayed the rise of pH, Hue angle, and moisture values of chilled beef with time; slowed down the lipid oxidation and the growth of surface microorganisms of chilled beef. Altogether, the prepared biological nanocomposites can be used as promising materials to replace commercial and non-degradable plastic films.

Comment 3: In the last paragraph of the introduction, the Author needs to clearly state the novelty of this paper together with future prospects of this study.

Response 3: Thank you very much. We revised the last paragraph of introduction, and clearly state the novelty of present manuscript. The revised paragraph was as follow:

In the previous study, we have reported the antibacterial activity and mechanism of GEO against E. coli and Staphylococcus aureus [18]. In present research, the polysaccharide chitosan as the polymer matrix was combined with nano-clay montmorillonite and natural active substance GEO to prepare GEO, chitosan, and montmorillonite novel bio-nano composite films. The composite films were tested for antioxidant and antibacterial properties, and their sustained-release performance, safety, and stability attributes were also evaluated. Our results demonstrated that the resulting bio-nano films not only overcomed the problem of auto-oxidation of GEO but they could also improve the properties of chitosan film. Therefore, the prepared biological nanocomposites can be used as promising materials to replace commercial and non-degradable plastic films.

Comment 4: Authors need to follow the journal format fully in the case of the Reference list. For example, Journal abbreviations, heading, and subheadings etc.

Response 4: Thank you very much. We edited the manuscript follow the journal format.

Comment 5: In the result and discussion section, the author needs to pay more attention and validate their findings with recent previous results and compare if possible.

Response 5: Thank you very much. We paid more attention and validated our results with recent previous results and compare.

Comment 6: All Figure caption needs to be updated including detailed experimental conditions as precisely as possible. The presentation of the Figure needs to be improved.

Response 6: Thank you very much.

We updated the Figure captions, which include the detailed experimental conditions.

We changed Figure 2, and improved the solution of image. Figure captions were revised as follow:

Figure 1. Fourier Transform infrared spectra of (Black) Ch, (Orange) Ch+MMT, (Blue) Ch+MMT+0.1 % GEO, (Pink) Ch+MMT+0.3% GEO, (Green) Ch+MMT+0.5 % GEO. The spectra data were obtained in the range of 4000 to 400 cm−1 at the resolution of 4 cm−1, and with 56 scans per spectrum.

Figure 2. Migration of active components in different kinds of fresh-keeping films. The migration of active components was determined using the Folin phenol method, which was determined by the content of the phenolic compounds that migrated into 95% ethanol solution at 37 °C.

Comment 7: The conclusion part needs to be revised and English language usage mistakes must be removed.

Response 7: Thank you very much. We revised conclusion part, and invited Dr. Kannan RR Rengasamy from University of Limpopo to revise the gramma.

Reviewer 2 Report

The manuscript entitled "Preparation and characterization of bio-nano composites film of chitosan and montmorillonite incorporated with ginger essential oil and its application in chilled beef preservation" described the incorporation of ginger essential oils tp a biofilms made with chitosan and montmorillonite. The manuscript is well-written, the results are clear and well presented.  In the Figure 1, in the  stacked IR, seems like when the percentage of GEO increased the signal at 1550 cm-1 disappeared, please discussed about it. My main criticism is the lack of discussion and comparison of other biofilms with the same application. and highlight the superior applicability to its biofilms. 

Good manuscript after address my comments, it will be accepted with minor changes.

Author Response

The manuscript entitled "Preparation and characterization of bio-nano composites film of chitosan and montmorillonite incorporated with ginger essential oil and its application in chilled beef preservation" described the incorporation of ginger essential oils to a biofilms made with chitosan and montmorillonite. The manuscript is well-written, the results are clear and well presented. 

Comment 1:  In the Figure 1, in the stacked IR, seems like when the percentage of GEO increased the signal at 1550 cm-1 disappeared, please discussed about it.

Response 1: Thank you very much for pointing this interesting point. We think the signal at 1550 cm-1 almost disappeared with the percentage of GEO increased, which might due to the amide group reaction with GEO to form new functional groups. However, we cannot decide whatwas the exact groups now. We will do this interesting work in our next research.

Comment 2:  My main criticism is the lack of discussion and comparison of other biofilms with the same application. and highlight the superior applicability to its biofilms. 

Response 2: Thank you very much. Your suggestion is very fine.

In revised manuscript, we highlighted the superior applicability of our prepared biofilms, e.g., To summarize, the bio-materials have demonstrated effective preservation effects, and the nanocomposite films have been identified as potential preservative films, effectively extending the shelf life of meat. The bio-nano composite films containing different proportions of ginger essential oil (GEO) might be convenient and effective technology for beef preservation.

Comment 3:  Good manuscript after addresses my comments, it will be accepted with minor changes.

Response: Thank you very much.

Reviewer 3 Report

The authors have performed the preparation and characterization of bio-nano composites film of chitosan and montmorillonite incorporated with ginger essential oil and its application in chilled beef preservation. The work is technical sound and the authors utilized appropriate techniques of analysis. The results are supported by the data and supply useful conclusion. There are some typewriting errors and some sentences are rambling. The conclusion section must be improved to better explain the obtained results and their potentiality:

I only have a few comments:

- In the abstract, the mean of the sentence “In this study; bio-nanocomposite films containing different proportions of ginger essential oil (GEO), Chitosan (Ch), and Montmorillonite (MMT) were prepared and characterized; and the preservation effect of bio-nano composite films on chilled beef was evaluated.” Is not clear. Please reformulated it.

- In the abstract, the sentence “In the process of preservation of the bio-nano composite films; Ch +0.5% GEO group; and Ch +MMT+0.5% GEO group effectively delayed the rise of pH; Hue angle; and moisture values of chilled beef with time; slowed down the lipid oxidation and the growth of surface microorganisms of chilled beef; maintaining the quality of chilled beef; extended the shelf life.” Is too long. Please divide into two or more sentences.

-In figure 2: please increase the quality of the figure. It is hard to read along the axis and also the corresponding title.

-Page 6 line 229, 232, 246: please change 15th with 15th.

Author Response

The authors have performed the preparation and characterization of bio-nano composites film of chitosan and montmorillonite incorporated with ginger essential oil and its application in chilled beef preservation. The work is technical sound and the authors utilized appropriate techniques of analysis. The results are supported by the data and supply useful conclusion. There are some typewriting errors and some sentences are rambling. The conclusion section must be improved to better explain the obtained results and their potentiality:

I only have a few comments:

Comment 1- In the abstract, the mean of the sentence “In this study; bio-nanocomposite films containing different proportions of ginger essential oil (GEO), Chitosan (Ch), and Montmorillonite (MMT) were prepared and characterized; and the preservation effect of bio-nano composite films on chilled beef was evaluated.” Is not clear. Please reformulated it.

Response 1: Thank you very much. Your suggestion is reasonable.

We revised this sentence as ‘In this study, bio-nano composite films containing different proportions of ginger essential oil (GEO), Chitosan (Ch), and Montmorillonite (MMT) were prepared and characterized, and the antibacterial effect of bio-nano composite films on chilled beef was evaluated.’ in revised text.

Comment 2- In the abstract, the sentence “In the process of preservation of the bio-nano composite films; Ch +0.5% GEO group; and Ch +MMT+0.5% GEO group effectively delayed the rise of pH; Hue angle; and moisture values of chilled beef with time; slowed down the lipid oxidation and the growth of surface microorganisms of chilled beef; maintaining the quality of chilled beef; extended the shelf life.” Is too long. Please divide into two or more sentences.

Response 1: Thank you very much. Your suggestion is reasonable.

We revised this sentence as ‘The bio-nano composite films (Ch +0.5% GEO group, and Ch +MMT+0.5% GEO group) effectively delayed the rise of pH, Hue angle, and moisture values of chilled beef with time; slowed down the lipid oxidation and the growth of surface microorganisms of chilled beef.’ in revised text.

Comment 3 -In figure 2: please increase the quality of the figure. It is hard to read along the axis and also the corresponding title.

Response 3: Thank you very much.

We changed Fig. 2 with 600 dpi in revised manuscript.

Comment 4 -Page 6 line 229, 232, 246: please change 15th with 15th.

Response 4: Thank you very much. We changed it in revised manuscript.